# Faster Walking Speeds Require Greater Activity from the Primary Motor Cortex in Older Adults Compared to Younger Adults

**DOI:** 10.3390/s23156921

**Published:** 2023-08-03

**Authors:** Lisa Alcock, Rodrigo Vitório, Samuel Stuart, Lynn Rochester, Annette Pantall

**Affiliations:** 1Translational and Clinical Research Institute, Newcastle University, Newcastle upon Tyne NE4 5PL, UK; lisa.alcock@newcastle.ac.uk (L.A.); lynn.rochester@newcastle.ac.uk (L.R.); 2National Institute for Health and Care Research (NIHR), Newcastle Biomedical Research Centre (BRC), Newcastle University, Newcastle upon Tyne NE4 5PL, UK; 3Department of Sport, Exercise and Rehabilitation, Northumbria University, Newcastle upon Tyne NE1 8ST, UK; rodrigo.vitorio@northumbria.ac.uk (R.V.); samuel2.stuart@northumbria.ac.uk (S.S.); 4Department of Neurology, Oregon Health & Science University, Portland, OR 97239, USA; 5The Newcastle upon Tyne NHS Foundation Trust, Newcastle upon Tyne NE1 4LP, UK; 6Population Health Sciences Institute, Newcastle University, Newcastle upon Tyne NE2 4AX, UK

**Keywords:** functional near infrared spectroscopy, oxygenated haemoglobin, frontal lobe, cortex, preferred gait velocity, fast gait velocity

## Abstract

Gait speed declines with age and slower walking speeds are associated with poor health outcomes. Understanding why we do not walk faster as we age, despite being able to, has implications for rehabilitation. Changes in regional oxygenated haemoglobin (HbO2) across the frontal lobe were monitored using functional near infrared spectroscopy in 17 young and 18 older adults while they walked on a treadmill for 5 min, alternating between 30 s of walking at a preferred and fast (120% preferred) speed. Gait was quantified using a triaxial accelerometer (lower back). Differences between task (preferred/fast) and group (young/old) and associations between regional HbO2 and gait were evaluated. Paired tests indicated increased HbO2 in the supplementary motor area (right) and primary motor cortex (left and right) in older adults when walking fast (*p* < 0.006). HbO2 did not significantly change in the young when walking fast, despite both groups modulating gait. When evaluating the effect of age (linear mixed effects model), greater increases in HbO2 were observed for older adults when walking fast (prefrontal cortex, premotor cortex, supplementary motor area and primary motor cortex) compared to young adults. In older adults, increased step length and reduced step length variability were associated with larger increases in HbO2 across multiple regions when walking fast. Walking fast required increased activation of motor regions in older adults, which may serve as a therapeutic target for rehabilitation. Widespread increases in HbO2 across the frontal cortex highlight that walking fast represents a resource-intensive task as we age.

## 1. Introduction

Complex neural processes underpin bipedal locomotion, with cortical, subcortical and spinal structures involved [1,2,3]. Mobile neuroimaging techniques such as functional near infrared spectroscopy (fNIRS) and electroencephalography permit the investigation of the underlying neural control of movement across the cortex during actual gait, as opposed to other neuroimaging modalities (i.e., magnetic resonance imaging) which often use constrained protocols (such as asking an individual to remain still while imagining movement or performing repeated contractions of a single joint [4]) or explore cross-sectional and longitudinal associations between neuroanatomical function and gait [5,6]. Collectively, these approaches have generated an understanding of frontal lobe function and its contribution to locomotion.

Constituting the largest lobe in the brain, the frontal lobe is reportedly the most susceptible region to age-related atrophy [7,8] and comprises cortical regions involved in the execution of voluntary movements including gait. The prefrontal cortex is linked with attention, planning and executive functions [9,10] and is involved in complex gait tasks that require *increased attention* (i.e., dual tasking [11,12,13,14], prepared walking [15]), *greater motor demand* (i.e., increasing locomotor speed [16]) or *challenge balance* (i.e., negotiating an obstacle [17], responding to an anticipated postural perturbation [18]). The premotor cortex [15] and supplementary motor area [19,20] both contribute to the selection, planning and co-ordination of voluntary movements including gait. Evidence from primate studies indicates that the premotor cortex governs externally guided movements whereas the supplementary motor area is responsible for internally guided movements [21]. The primary motor cortex is located anteriorly in the precentral gyrus and controls muscle activation and movement through direct connections to spinal motor neurons and also receives inputs from other motor regions, including cortical (premotor cortex, supplementary motor area and parietal cortex) and subcortical (basal ganglia, cerebellum) structures.

The speed at which we walk represents a measure of gross motor function and overall health and well-being. As such, gait speed is recognised as a vital clinical sign [22,23,24], alongside other physiological measures including heart rate, respiratory rate, blood pressure, temperature and pain. Slower walking speeds are associated with a number of clinical outcomes including disability [25], cognitive impairment [26], hospitalization [27], institutionalisation and falls [28]. Age-related changes in walking speed are further exacerbated in the presence of age-related neurodegenerative conditions [29] (such as Parkinson’s disease [30] and dementia [31]), age-related comorbidities (i.e., arthritis, sarcopenia and osteoporosis [32], frailty [33], hypertension [34,35] and obesity [36]), polypharmacy [37], depression and mood [38,39,40]. Preferred walking speed reduces with age [41], particularly after 80 years [42], and is predictive of life expectancy and mortality (individuals who walk slower tend to live shorter lives) [43,44,45].

Understanding why older adults do not walk faster, despite being able to, has implications for healthcare services and rehabilitation. There have been limited studies investigating age-related changes in regional cortical activity when walking fast. The prefrontal cortex and premotor cortex have been implicated in the modulation of gait speed in young adults, particularly when transitioning from walking to running, with limited involvement from the sensorimotor cortices [16]. However, in older adults, increased cortical activation (increased oxygenated haemoglobin; HbO2) has been observed in the dorsolateral prefrontal cortex and supplementary motor area [46,47] when walking overground at a self-selected slow, preferred and fast speed compared to young adults [47] and when walking was evaluated at 30%, 50% and 70% intensity in older adults only [46]. The relationship between gait and cognition in older age is complex [48]. Gait is considered a cognitively demanding task [47], particularly in older adults, due to attentional and resource allocation deficits affecting task prioritisation [49], reduced cognitive capacity and mobility limitations [50]. This is affirmed by studies observing increased prefrontal cortex activity in older adults compared to young adults during overground walking [47,51]. Our understanding of age-related cortical changes across multiple regions simultaneously when walking at a preferred and fast speed suggests increased cortical activity in young and older adults when increasing speed. However, no previous study has investigated the role of the primary motor cortex and age-related changes when walking at preferred and fast speeds.

This study investigates age-related changes in regional cortical activity across the frontal cortex and temporal-spatial gait outcomes when walking at a preferred and fast speed. The aims of this study were to: (1) evaluate the change in regional cortical activity and gait in response to walking fast (120% preferred walking speed) and investigate differences between young and old adults; and (2) explore associations between regional cortical activity and temporal-spatial gait outcomes. We hypothesised that cortical activity would be higher in older adults compared to young adults across all regions, particularly the prefrontal cortex, premotor cortex and supplementary motor area, as well as the primary motor cortex, with increased cortical activity observed during fast walking. We propose that increased cortical activity would be related to gait outcomes pertaining to forward progression (step length, step time, swing time) and reduced gait variability in older adults.

## 2. Materials and Methods

### 2.1. Participants

Pilot data collected from healthy young and older adults were used to inform power calculations which indicated that a sample size of 30 (15 per group) was required to identify a change of 4.3% with a power of 0.8. A convenience sample of 17 young (20–40 years) and 18 older (≥60 years) adults were recruited through advertisement and community partnerships [13,52]. Additional participants were recruited to account for potential technical difficulties during data acquisition. All participants were independently mobile, community dwelling and had adequate hearing (correct recall using the Whisper test [53]) and vision (attaining a visual acuity score of 6/12 on the Snellen chart [54]). Participants were excluded if they reported gait abnormality, severe depression or had a clinical diagnosis of dementia (score < 21 on the Montreal Cognitive Assessment; MOCA [55]) or other neurological condition, musculoskeletal disorder affecting the lower limbs, cardiovascular or respiratory disease or chronic pain. Ethical approval was granted from the institutional ethics review board and all participants provided written, informed consent. Participants completed all assessments during a single session lasting 2–3 h. Demographic (age, sex), anthropometric (height, mass and body mass index; BMI) and clinical (fear of falling using the International falls efficacy scale; FES-I [score/64] [56], global cognition using the MOCA [score/30]) descriptors were obtained. All participants were right-handed except for one older male.

### 2.2. Study Design, Apparatus and Data Acquisition

Participants completed a timed overground walk covering a 10 m distance for quantifying preferred walking speed. A warm-up bout of up to 2 min walking was completed on the treadmill prior to data collection to ensure participants were familiarised with the experimental procedures. None of the participants reported difficulty walking on the treadmill. Participants completed five minutes walking on the treadmill, alternating between 30 s at preferred speed and 30 s at a fast speed (120% of preferred speed). A block design with five repetitions of control (preferred speed) and experimental (fast speed) manipulations was employed [57] to avoid anticipatory responses [15]. Usual arm swing was permitted. Specific procedures for acquisition of fNIRS and gait are described below and are reported in previously published studies [13,52].

#### 2.2.1. Regional HbO2

A tethered functional near-infrared spectroscopy (fNIRS) system (LABNIRS, Shimadzu, Kyoto Japan) was used to monitor changes in total haemoglobin, oxygenated haemoglobin (HbO2) and deoxygenated haemoglobin (HHb) using continuous wave laser diodes with wavelengths of 780 nm, 805 nm and 830 nm sampling at 23.8 Hz. The modified Beer–Lambert Law estimated the concentration of chromophores HbO2 and HHb through calculating the attenuation of near-infrared light at different wavelengths [58]. The interoptode distance was 30/35 mm. Data were recorded from 40 channels (25 optodes; 5 × 5, 13 transmitters and 12 detectors) covering both the right and left hemisphere of the frontal lobe. Participants wore a whole-head fiber holder cap configured according to the international 10-10 electroencephalography system (Shimadzu, Kyoto, Japan). Three dimensional co-ordinates of anatomical reference locations (Cz, nasion, left and right preauricular) and optode locations were ascertained using a digitizer (FASTRAK, Polhemus, VT, USA). An open-source spatial registration routine implemented in NIRS-SPM v4_r1 (http://www.nitrc.org/projects/nirs_spm accessed on 2 February 2017) [59] was used to calibrate signals recorded from the cortices according to the digitised scalp locations. fNIRS channels were organised according to the Montreal Neurological Institute standard space [60]. Cortical regions included the prefrontal cortex (Brodmann areas 8, 9, 10, 45, 46), premotor cortex (Brodmann area 6, lateral), supplementary motor area (Brodmann area 6, medial) and primary motor cortex (Brodmann area 4).

#### 2.2.2. Gait

Participants wore a small, lightweight triaxial accelerometer (Axivity AX3, York, UK; dimensions H × W × D: 32.5 × 23.0 × 7.6 mm, 9 g) on the lower back at the fifth lumbar vertebrae (L5). The accelerometer was affixed to the skin surface using double-sided adhesive and secured in place using a Hypafix plaster (BSN Medical Limited, Hull, UK). The device measures continuous acceleration signals in the vertical, anterior-posterior and mediolateral directions sampling at 100 Hz (resolution: 16-bit, range: ±8 g).

### 2.3. Data Analysis

#### 2.3.1. Regional HbO2

Procedures for analysis of fNIRS data were completed in NIRS-SPM and followed recommended guidance where possible [57,61]. The HbO2 signal was selected for analysis as it is more sensitive to task-related changes [16,20,62]. HbO2 signals were preprocessed using the Time Series Analysis routine which included signal filtering, detrending and baseline correction. A low-pass filter with a cut-off frequency of 0.15 Hz was used to remove high-frequency noise assuming a canonical hemodynamic response function [63]. HbO2 signals were decomposed into global trends (arising from biological signals attributed to respiration, cardiac function and movement artefacts), hemodynamic signals and unrelated noise using a wavelet-minimum description length detrending algorithm [64]. Regional HbO2 were further analysed in MATLAB^®^ (R2015a, Mathworks, Inc., Natick, MA, USA). Channels were organised and averaged across each of the 8 cortical regions of interest, dividing them by the maximum signal amplitude recorded over the 5 min walk (%) [65]. For each 30 s trial, the first and last 5 s were discarded to remove signal changes resultant from anticipatory responses associated with transitioning between tasks (preferred and fast walking speeds) [16] as well as to account for the 4–6 s delay associated with the hemodynamic response [66]. For each 30 s bout of preferred walking, data recorded from 10–15 s were used for signal normalisation and data from 15–25 s were considered to reflect HbO2 associated with preferred walking. For each 30 s bout of fast walking, data from 5–25 s were extracted and evaluated over early (5–15 s) and late (15–25 s) phases.

#### 2.3.2. Gait

Custom algorithms were used to process accelerometer data and derive 8 temporal-spatial gait outcomes. Specifically, continuous wavelet transformations identified gait events (initial and final foot contacts) to segment the signal into gait cycles for estimating temporal variables. An inverted pendulum model was assumed for the estimation of spatial variables [67]. This pseudocode for gait feature extraction has been published [68] and these methods have been validated and are suitable for use with young and older adults [69]. Mean step length (m), step time (seconds), stance time (seconds) and swing time (seconds) were extracted. Gait variability (standard deviation of all steps) was computed for each gait outcome.

### 2.4. Statistical Analysis

All statistical analyses were completed in SPSS (IBM, Armonk, NY, USA, Release 25.0.0.1). Data distributions were checked for normality and confirmed by skewness and/or kurtosis > 1.96 [70]. BMI and fear of falling were not normally distributed when stratified by age. Group differences in demographic, anthropometric and clinical outcomes were evaluated using independent *t*-tests, Mann–Whitney U tests and Chi squared, as appropriate.

Paired comparisons evaluated the differences in regional HbO2 between early (5–15 s) and late (15–25 s) phases of the fast walking trials (total comparisons n = 40/age group; 5 trials × 8 regions) using Mann–Whitney U tests. The majority of comparisons were not statistically significant (*p* ≥ 0.05) for either the young (n = 32/40, 80%) or older adult (n = 39/40, 98%) groups. As such, the early and late phases for the fast walking trials were collapsed.

#### 2.4.1. Aim 1—Change in HbO2 and Gait When Walking at a Preferred and Fast Speed

HbO2 and mean gait variables were normally distributed; thus, parametric statistics were used (paired *t*-test, data reported as mean [SD]). Gait variability was not normally distributed; thus, non-parametric statistics were used (Wilcoxon signed ranks test, data presented as median [25th, 75th percentile]).

Within-group differences in regional HbO2 and gait (absolute values) were evaluated by comparing preferred and fast walking. Statistical comparisons were conducted for young and older adults independently using independent *t*-tests and Mann–Whitney U as appropriate. A Bonferroni adjustment corrected for multiple comparisons with statistical significance accepted when *p* < 0.05/comparisons (n). When significant differences were identified, effect sizes were calculated [71]. For normally distributed outcomes, Hedges g was calculated (Equation (1)) as opposed to Cohen’s d to avoid nonnegligible bias for small sample sizes. For outcomes that were not normally distributed, effect sizes were also calculated using the z statistic from the Wilcoxon signed ranks test (Equation (2)). Effect sizes ranged from 0–1 and were interpreted as large (0.5 ≥ 1.0), medium (0.3 ≥ 0.5) and small (0.1 > 0.3) [72].
g = MFast − MPreferred/s(1)

M denotes mean of the fast (MFast) and preferred (MPreferred) walking speed conditions, s denotes the pooled sample standard deviation.

r = z/SQRT(N)(2)

z denotes the z statistic derived from the Wilcoxon signed ranks test, SQRT denotes square root and N denotes the total samples.

Radar plots were used for visualising changes in regional HbO2 and gait (absolute data) outcomes. For these plots, data were transformed to z scores (Equation (3)), permitting outcomes with different units of measurement to be plotted on the same graph and scale and reviewed collectively.

z score = (x − M)/sd(3)

x is the data value, M the group mean and sd the standard deviation.

To enable within- and between-group differences to be evaluated in the same linear mixed effects model (LMEM), difference scores were calculated for regional HbO2 and gait (∆ fast − preferred). Difference scores were not normally distributed; however, LMEMs are considered robust to small normality violations [73]. A LMEM was ran for HbO2 (per region) and each of the gait outcomes to determine the effect of age and trial (fixed factor: group and trial). Demographic, anthropometric and clinical descriptors that significantly differed between the groups (*p* ≤ 0.05) or trended towards significance (*p* ≤ 0.10), except for age, were included as covariates. Considering the differences in brain structure and function [74,75] and cerebral hemodynamics [76] in males vs. females, sex was entered as a fixed factor and interaction effect (group × sex) of no interest. Box plots are utilised to transparently display the distribution of difference scores (∆ fast − preferred).

#### 2.4.2. Aim 2—Association between Regional HbO2 and Gait in Young and Older Adults

All correlations were computed for young and old adults independently. Bivariate correlations (Spearman’s rho) were used to evaluate associations between regional HbO2 (∆ fast − preferred), gait (∆ fast − preferred) and group descriptors that were included in the LMEM as covariates.

Partial correlations (Spearman’s rho) evaluated associations between difference scores (∆ fast − preferred) in regional HbO2 and gait. Group descriptors that were controlled for in the LMEM were included as covariates in partial correlations between regional HbO2 and gait outcomes. Correlation strength was interpreted as weak (rho < 0.40), moderate (0.40 ≤ rho < 0.70) or strong (rho ≥ 0.70). For bivariate and partial correlations, statistical significance was accepted *p* ≤ 0.05. To aid visual interpretation of the bivariate and partial correlations and identify patterns, a correlation matrix will highlight the relationship direction (positive or negative) using colour and vary colour intensity to demonstrate correlation strength.

## 3. Results

The groups were well-matched in proportion of males to females, height, mass, global cognition (MOCA) and fear of falling (FES-I) (*p* > 0.05, Table 1). Older adults walked significantly slower than young adults (*p* ≤ 0.001) and there was a trend for the older adults to have a higher BMI (*p* = 0.072).

### 3.1. Changes in Regional HbO2 Concentration and Gait When Walking at a Preferred and Fast Speed (Aim 1)

Using absolute cortical activity (regional HbO2; %) and gait, changes that occurred when walking fast were evaluated using paired comparisons in the young and older adult groups independently.

#### 3.1.1. Regional HbO2 in Young and Older Adults

Regional HbO2 did not significantly change from preferred to fast walking in young adults (*p* > 0.05; Table 2, Figure 1). A significant increase in HbO2 was noted during fast walking across multiple motor regions (right supplementary motor area, left and right primary motor cortex, *p* ≤ 0.006) compared to preferred walking in older adults (*p* ≤ 0.006). Effect sizes (g) indicated a medium-sized effect, with the largest difference noted for the right supplementary motor area (g = 0.48). For the remaining five regions, group mean HbO2 increased, although these differences were not considered significant post-Bonferroni correction (*p* ≤ 0.006).

#### 3.1.2. Gait in Young and Older Adults

Five of the eight gait outcomes were significantly different between the two walking speed conditions in the young adults. When walking fast, the young adults walked with a reduced step time, stance time and swing time (*p* ≤ 0.001), increased step length (*p* ≤ 0.001) and reduced step length variability compared to when walking at a preferred speed (Table 3, Figure 1). Effect sizes indicated moderate (step length g = 0.35) to strong (step time, stance time, swing time and step length variability g/r > 0.60) differences.

Fewer adaptations in gait were observed for older adults (three of the eight gait outcomes). When walking fast, older adults walked with a reduced step time and swing time (*p* ≤ 0.005) as well as reduced step length variability (*p* = 0.006) compared to when walking at a preferred speed (Table 3, Figure 1). Effect sizes indicated moderate differences in step time, swing time and step length variability (g/r = 0.42–0.58).

LMEMs were used to compare changes in regional HbO2 and gait between groups using difference scores. There was no main effect of the trial on regional HbO2 or gait outcomes, so this was removed as a parameter of interest and the models were re-run (Appendix A Appendix A).

#### 3.1.3. Influence of Age on Regional HbO2 and Gait

The LMEM revealed a main effect of group (age) on HbO2 across five regions (left prefrontal cortex, left premotor cortex, supplementary motor area (left and right) and the left primary motor cortex), indicating larger increases in HbO2 during fast walking for older adults compared to young adults (Figure 2). At least half of the older adults increased HbO2 in the left prefrontal cortex (50%, n = 9/18), left premotor cortex (50%, n = 9/18), left supplementary motor area (61%, n = 11/18), right supplementary motor area (83%, n = 15/18) and left primary motor cortex (67%, n = 12/18) when walking fast.

The LMEM revealed a main effect of group (age) on gait (Figure 3). Temporal gait outcomes (step time, stance time and swing time) were reduced during fast walking in both groups (indicated by negative values). Step length was longer in both groups during fast walking (indicated by positive values). While temporal gait variability (step time, stance time and swing time) was higher in older adults during preferred walking (*p* ≤ 0.024), only stance time variability was considered statistically significant post-Bonferroni correction (*p* ≤ 0.006).

### 3.2. Associations between Regional HbO2 and Gait (Aim 2)

#### 3.2.1. Bivariate Correlations

Bivariate correlations (rho) were computed to evaluate the relationship between descriptor outcomes that differed between young and old adults (BMI and 10 m overground speed) and regional cortical activity and gait outcomes using difference scores (∆ fast − preferred). No significant correlations were found between regional HbO2 and gait for the young adults (Table 4).

Positive correlations of moderate strength (>0.4) were identified for 10 m overground walk speed and HbO2 in the prefrontal cortex (left and right) and left premotor cortex in older adults. Only the relationship between the left prefrontal cortex and 10 m overground walk speed was statistically significant (Figure 4; rho = 0.48, *p* ≤ 0.05). Negative correlations of moderate strength (>−0.4) were observed for 10 m overground walk speed and BMI with gait in older adults. Slower overground walking speeds were associated with smaller differences in step time and stance time when walking fast compared to a preferred speed. Similarly, a lower BMI was associated with smaller differences in gait variability (swing time and step length) when walking fast compared to preferred speed in older adults; however, only the association between BMI and swing time variability was statistically significant (Figure 4; rho = −0.47, *p* ≤ 0.05).

#### 3.2.2. Partial Correlations

Partial correlations were computed for young and old adults separately using difference scores (∆ fast − preferred) and controlling for BMI and preferred gait speed (Table 4). When interpreting correlations involving difference scores, it is important to consider the distribution of data values across the quadrants reviewed through the visual observation of xy scatterplots in addition to the strength and significance of the correlation, as each reflects a different relationship (positive/negative x values vs. positive/negative y values).

Controlling for 10 m overground walk speed and BMI, positive correlations of moderate strength were observed in young adults for the right prefrontal cortex with step length and step length variability; however, neither of these relationships were statistically significant. In older adults, a greater number of moderate to strong correlations were identified. In particular, correlations existed between step length and step length variability with regional HbO2. Positive correlations of moderate strength were identified for step length and HbO2 in the prefrontal cortex (left: rho = 0.65, *p* ≤ 0.01 and right: rho = 0.62, *p* ≤ 0.01), right premotor cortex (rho = 0.73, *p* ≤ 0.001), supplementary motor area (left: rho = 0.62, *p* ≤ 0.01 and right: rho = 0.47, non sig.) and primary motor cortex (left: rho = 0.46, non sig. and right: rho = 0.53, *p* ≤ 0.05). Positive correlations of moderate strength were noted for step length variability with HbO2 in the prefrontal cortex (left and right) and right supplementary motor area (rho range = 0.45–0.51); however, none were statistically significant. Only correlations between step length variability and HbO2 in the right premotor cortex (rho = 0.70) and right primary motor cortex (rho = 0.51) were significant (*p* ≤ 0.05). Significant partial correlations are plotted for step length (Figure 5) and step length variability (Figure 6). The majority of older adults increased step length when walking fast (upper right quadrant in Figure 5), and this was associated with increased HbO2 in the prefrontal cortex (left and right), right premotor cortex, left supplementary motor area and right primary motor cortex (upper right quadrants in Figure 5). The majority of older adults reduced step length variability when walking fast, and this was associated with increased HbO2 in the right premotor cortex and left supplementary motor area (upper left quadrant in Figure 6).

## 4. Discussion

Gait is considered a complex task, integrating cognitive and multi-sensory (visual, somatosensory and vestibular) inputs to inform locomotor patterns and adaptations. This study has demonstrated that whilst young adults modulate walking faster by making a greater number of gait adaptations, older adults required greater cortical activation in motor regions, particularly the right supplementary motor area and primary motor cortex (right and left). When investigating the influence of age upon differences in regional HbO2 and gait, difference scores (∆ fast − preferred) were evaluated while controlling for group differences in 10 m overground walk speed and anthropometric characteristics (sex and BMI). Compared to young adults, greater increases in HbO2 were observed in older adults across a number of regions: the left prefrontal cortex, left premotor cortex, supplementary motor area (left and right) and left primary motor cortex.

### 4.1. Changes in Regional HbO2 and Gait: Preferred vs. Fast Speed

By comparing treadmill walking speed conditions in young and older adults independently, each participant served as their own control, negating the need for covariates. The young adults made a greater number of gait adaptations to walk fast compared to older adults. Whilst the relative change in walking speed was standardised (i.e., 20% of preferred walking speed), the absolute increase in walking speed was greater for the young, as they walked significantly faster during the 10 m overground walk. Despite this, regional HbO2 remained stable (no significant changes) in the young adults in response to walking fast. This likely reflects a lower task demand, higher cognitive reserve and effective cortical resource allocation in response to faster walking speeds in young adults [77]. In the present study, significant increases in HbO2 were noted across the primary motor cortex (left and right) in older adults when walking fast, demonstrating the role of the primary motor cortex in increasing walking speed.

Walking speed is often used as a primary outcome in clinical trials, and normative databases are available from which normative thresholds have been developed [42]. Overground gait speeds were considered slow in both the young (<1.3 m/s) and older adult (<1.0 m/s) groups. For example, walking speeds of <1.0 m/s are often associated with poor health, reduced function [27] and dependence [28]; however, the participants included in the present study were living independently in the community, were ambulatory and reported no known health concerns. Slow overground walking speeds may be explained by the protocol used for measuring preferred speed, which involved participants traversing overground from a standing start and recording the time taken to complete a distance of 10 m using a stopwatch. While gait speed can be measured easily without sophisticated equipment and without the need for specialist expertise [29,78], calculations for preferred 10 m overground speed (distance divided by time) assume that velocity is constant (i.e., for every unit of time, the same distance is covered). However, the 10 m overground walk test in the present study incorporated both acceleration and deceleration phases and is not representative of steady-state walking speed. Despite the slower walking speeds, regional HbO2 was significantly increased in older adults and significant gait adaptations were observed in both groups in response to walking fast, suggesting that the change in gait speed from preferred to fast was of sufficient challenge. Investigating cortical activity during preferred and fast walking may be relevant for many other conditions where major systems are affected and mobility is reduced (reflected by slower gait speeds), including individuals with poor neurosensory function, muscle function, cardiometabolic function and adiposity, which are all predictive of walking speed [79].

### 4.2. Changes in Regional HbO2 and Gait: Young vs. Older Adults

Overactivation in regional HbO2 across multiple regions in the frontal lobe may be an indication of neural compensation in the older adults. In response to increased task complexity, greater neural resources were required in older adults. A number of theories exist to understand changes in brain activity with older age. The Hemispheric Asymmetry Reduction in Older Adults (HAROLD) model suggests that activation is unilateral in the young and bilateral in older age [80]. The Posterior-Anterior Shift in Aging (PASA) model theorises that posterior lobes are under-activated in favour of overactivation in the frontal lobe in older age [81]. The findings of the present study indicated increased cortical activity bilaterally in the supplementary motor area in older adults when walking fast (in agreement with HAROLD); however, this was not replicated in other areas (prefrontal cortex, premotor cortex, primary motor cortex). Daselaar and colleagues [82] tested the mechanism of overaction in older adults using event-related fMRI and neuropsychological tests. Their findings suggest that the age-related reduction in connectivity is compensated for by greater synaptic firing, which they propose is related to changes in white matter integrity (afferent fibres reduce in number but demonstrate increased firing potential per fibre).

It was anticipated that treadmill walking alone may serve as a complex cognitive task, further increasing in complexity when walking fast, even for the cognitively intact older adults recruited. Increased HbO2 in the prefrontal cortex in older adults when walking fast provides further support. Evidence from MRI studies suggests that white matter connections between the prefrontal cortex and subcortical areas are associated with gait stability [83]. Gait variability was reduced in older adults when walking fast, indicating a more rhythmical, stable gait pattern and suggesting that dynamic postural control was not compromised. As gait was measured with a single sensor on the lower back, it was not possible to estimate other proxy markers of postural control such as step width and step width variability.

### 4.3. Associations between Regional HbO2 and Gait in Young and Older Adults

Alterations in temporal gait outcomes (reduced step time, stance time and swing time) were observed in both groups when walking fast (Figure 1), which was not related to changes in cortical HbO2 in either group. It is likely that subcortical structures were responsible for the change in gait timing, in particular the cerebellum and associated circuity [84], as observed in patients with cerebellar ataxia who display deficits in timing and gait rhythmicity [85]. Moreover, it is considered that the constant speed of the treadmill may have reduced the inherent variability associated with overground walking and provided an external rhythmical cue, thereby underestimating the changes in cortical activity with reduced temporal gait variability. Our findings concur with previously published findings involving the same participants, indicating that reduced step length variability was associated with increased HbO2 across multiple regions bilaterally (premotor cortex, supplementary motor area and primary motor cortex) in older adults when walking with an auditory cue [52]. Similarly, increased HbO2 in the right premotor cortex and right primary motor cortex were associated with reduced step length variability when walking fast in the majority of older adults. External prompts such as auditory cues are designed to regulate the temporal rhythmicity and spatial configuration of walking, improving gait performance. Interactions between the basal ganglia and supplementary motor area are involved in the regulation of repetitive movements [86] and are considered to underpin locomotor control in addition to modulations resulting from sensory inputs [87]. Consequently, it is interesting to note that relationships of moderate strength (0.4–0.7), albeit not significant, were observed between step length variability and HbO2 in the prefrontal cortex (left and right) and right supplementary motor area in older adults. After further visualisation of the xy scatter plots (see Appendix A Appendix A), it was evident that the majority of older adults demonstrated increased HbO2 in the right prefrontal cortex and right supplementary motor area alongside reduced step length variability when walking fast. For the left prefrontal cortex, there was a mixed response; while the majority of older adults reduced step length variability when walking fast, concentration of HbO2 increased for some and was reduced for others. This suggests that walking fast on a treadmill was a challenging task for older adults (increased prefrontal cortex activity) that also served as an external prompt requiring input from the supplementary motor area.

Despite the spatial constraints imposed by the treadmill, moderate to strong correlations were identified for step length and step length variability with regional HbO2 in older adults. This in part supports our hypothesis that associations between regional HbO2 and gait outcomes pertaining to forward progression would exist. The Compensation-Related Utilization of Neural Circuits Hypothesis (CRUNCH) suggests that brain activity is closely related to task demand, such that overactivation is required when task demand increases to avoid detriments in task performance [88,89]. With increased task demand, neural resources are diminished earlier in older adults, which can result in reduced task performance. In support of this hypothesis, we identified significant associations between increased activation (HbO2) and improved task (gait) performance in older adults. Specifically, larger increases in HbO2 were associated with alterations in the spatial configuration of walking in older adults when walking fast (increased step length and reduced step length variability). In contrast, changes in regional HbO2 when walking fast were not significantly associated with changes in gait performance in young adults. We opted to standardise the challenge imposed when walking fast, increasing treadmill speed by a proportion of preferred walking speed overground (120%), which permitted investigation of increased task demand relative to individual capacity. It is likely that this change in walking speed was not sufficiently demanding to warrant increased cortical activation in the young adults. This is in agreement with previous findings demonstrating increased activation across frontal regions (prefrontal and premotor cortices) in young adults when transitioning from walking to running [16]. Moreover, increased cortical activation in response to increased task demand demonstrates adaptability, suggesting that the older adults in the present study were able to allocate cortical resources effectively and adapt gait patterns to meet the higher task demand of walking faster. Increasing the task demand further (i.e., >120% preferred speed) may have resulted in greater cortical capacity utilised and ineffective gait adaptations (i.e., increased gait variability). In other older adult populations (i.e., frail individuals or those with cognitive and/or motor impairments), cortical capacity may be further compromised and attentional resource allocation insufficient/ ineffective such that gait presents a highly challenging task with slower walking speeds required.

For the majority of older adults, reductions in step length variability were accompanied by increases in cortical activity when walking fast, suggesting that this element of gait control required greater cortical resource Taking a systems theory approach, reduced movement variability is considered to reflect a highly stable behaviour (Dynamic Systems Theory) that is related to reduced movement error, resulting in optimal accuracy and efficiency (Generalized Motor Program Theory) and requiring minimal input from the central nervous system (Uncontrolled Manifold Hypothesis) (for review see [90]). Gait variability is associated with vascular changes [91] and altered brain structure and function in older adults, in particular areas for sensorimotor integration and coordination (for review see [92]). In a clinical context, increased gait variability is associated with an increased incidence of disability [93], poor stability, cognitive impairment and an increased risk of falls [94,95]. Verghese and colleagues [96] demonstrated that the increased activation of the prefrontal cortex during a task challenging both cognitive and motor functions (dual task walking) may be a sensitive measure used to predict prospective fall risk in older adults. In a separate population-based study of 411 older adults, Callisaya and colleagues [97] showed that measures of gait (speed and cadence) and gait variability (step time, step length, double support phase) were associated with an increased risk of multiple falls recorded prospectively. Understanding the interplay between regional cortical activation and gait variability in prospective fallers would inform assessments for fall risk and personalised therapies for fall prevention.

### 4.4. Study Considerations

Previous studies have deployed a range of protocols involving both constrained (treadmill) and unconstrained (overground) gait when evaluating cortical activity at different walking speeds [16,46,47,51]. Both modalities have advantages and disadvantages. Treadmill walking poses a somewhat artificial mode of walking, with altered sensory inputs and contrasting environmental challenges compared to the real world. However, this protocol permitted a standardised manipulation of gait speed that accounted for individual differences (120% preferred walking speed) and the evaluation of multiple cortical regions using a tethered fNIRS system. It is noteworthy that increased cortical activity in the prefrontal cortex, premotor cortex and supplementary motor area has been observed in young adults when walking on a treadmill compared to overground walking [98], suggesting that treadmill walking poses alternative challenges to walking freely overground.

Using fNIRS, the cortical control of gait may be evaluated in real-time, enhancing our understanding of locomotor control. The tethered system used in the present study permitted observations from eight cortical regions of the frontal lobe simultaneously, which were referenced anatomically using a 3D digitiser. However, using such a system is limited to treadmill walking (as in the present study) or moving the system with the participant during the walking trials. Furthermore, equations used for estimating the path length vector corrected for age are only valid for adults aged ≤60 years and are fixed across the entire brain, which must be considered in light of the present findings [99].

Metabolic consumption is highest in the brain compared to all other organs and cerebral blood flow is critical for brain function. Age-related changes in cardiovascular function, such as increased stiffness and thickness of arterial walls, elevated systolic blood pressure and mean arterial pressure all increase resistance and impedance of blood flow [100]. Neurovascular coupling refers to the link between cerebral blood flow and the metabolic rate of oxygen consumption, with regions/structures with higher levels of oxygen utilisation receiving increased cerebral blood flow. In addition to age-related changes, walking fast may have been a physically demanding task particularly for older adults, resulting in exercise-related changes in systemic blood flow due to an increased demand for oxygen. Vital signs (including heart rate, blood pressure and temperature) and blood oxygen were not monitored during data collection, and this is a limitation of the present study. Synchronicity between cardiac rhythms and gait has been observed in young adults, particularly when walking at fast speeds [101]. Clinical- and research-grade wearable devices are widely available for measuring these changes dynamically (i.e., heart rate monitor, pulse oximeter) and should be monitored routinely in future fNIRS studies. Moreover, documenting a full medical history relating to respiratory conditions is recommended, as a slower gait speed and increased HbO2 in the prefrontal cortex were noted in older adults with a history of asthma [102].

### 4.5. Recommendations for Future Work

Findings from cross-sectional [103,104] and longitudinal [45] studies affirm a decline in gait speed with older age; however, when major gait decline occurs is unclear, with mixed reports of declining gait speed at 71 years [104], compared to others suggesting beyond 85 years [105]. The present findings suggest that the primary motor cortex is implicated in walking fast in older adults and may be a useful target for rehabilitation. In support, transcranial direct current (tDCS) stimulation applied to the primary motor cortex has been shown to positively impact gait, improving gait speed, gait variability and synchronicity [106,107] and postural control [108,109]. In addition, peripheral neuromodulation therapies show promise for improving mobility and gait function in older adults by modifying sensory inputs and promoting neuroplasticity [110]. Extending the present analysis to include a wider spectrum of older age would permit the cross-sectional evaluation of whether changes in regional HbO2 throughout older age are linear and whether the rate of change (line gradient) is constant or not. While the groups were well-matched for proportion of males to females, we opted to control for sex in the LMEM as a factor of no interest. Sex differences in cortical blood flow have been reported [111], and should be further explored in larger samples. Future efforts should collate common datasets to enable clustering techniques to be applied and changes/patterns related to age, sex and other anthropometrics investigated.

The present approach revealed insights concerning regional HbO2 when walking at preferred and fast speeds across the frontal cortex; thus, inferences regarding subcortical involvement from the basal ganglia and cerebellum were not possible, and this is a limitation of current mobile imaging technologies. Neuroimaging studies demonstrate that gait velocity is associated with brain health (grey matter atrophy and loss of white matter integrity) in older adults and can predict longitudinal changes in neuroanatomical structure and function [3]. Utilising multi-modal imaging (combination of neuroimaging and mobile imaging techniques) to discern mobility changes longitudinally in older age will address gaps in our current understanding of how age-related neuroanatomical changes in brain function influence cortical activation during locomotion and the mechanism(s) underpinning cortical reserve and compensation [112].

Using a single sensor on the lower back, it is possible to calculate gait asymmetry as the absolute difference between the right and left limbs [69]. However, this measure reflects the magnitude of asymmetry only and does not indicate whether asymmetry was larger on the right or the left side. Hemispheric differences in regional HbO2 were noted in older adults (i.e., bilateral changes in the supplementary motor area and changes in left primary motor cortex but not the right primary motor cortex; Figure 2). Further research should explore hemispheric differences in regional HbO2 and gait asymmetry.

## 5. Conclusions

Compared to walking at a preferred speed, walking fast required increased activation of motor regions (supplementary motor area and primary motor cortex) in older adults, which may serve as useful targets in rehabilitation. Age-related differences revealed widespread increases in cortical activity across the frontal cortex when walking fast, indicating that walking fast represents a resource-intensive task as we age. Modulations to the spatial configuration of walking (step length and step length variability) were associated with cortical activity across multiple regions in older adults and implicate the frontal cortex when sustaining fast walking speeds.

## Figures and Tables

**Figure 1 sensors-23-06921-f001:**
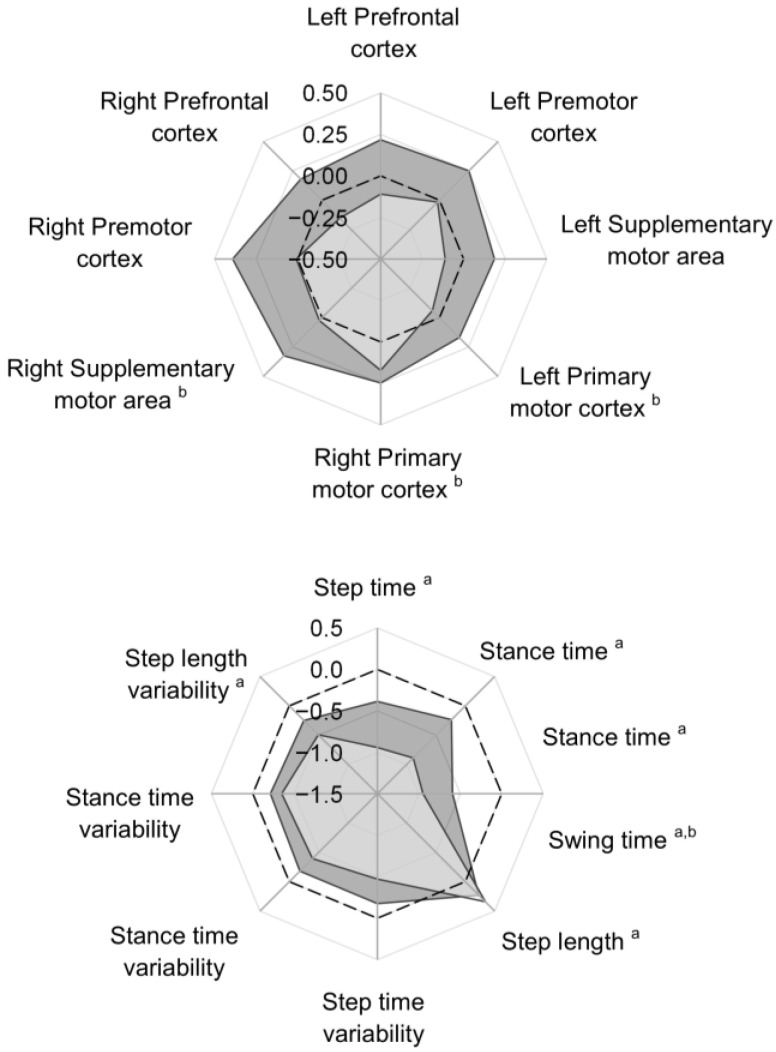
Radar plots demonstrating the change in regional HbO2 and gait for young (shaded light grey area) and older adults (shaded dark grey area) when walking fast compared to preferred speed (black dotted line at 0). Data are presented as z scores; ^a^ denotes significant differences between preferred and fast in young adults, ^b^ denotes significant differences between preferred and fast in older adults (post-Bonferroni correction: *p* ≤ 0.006).

**Figure 2 sensors-23-06921-f002:**
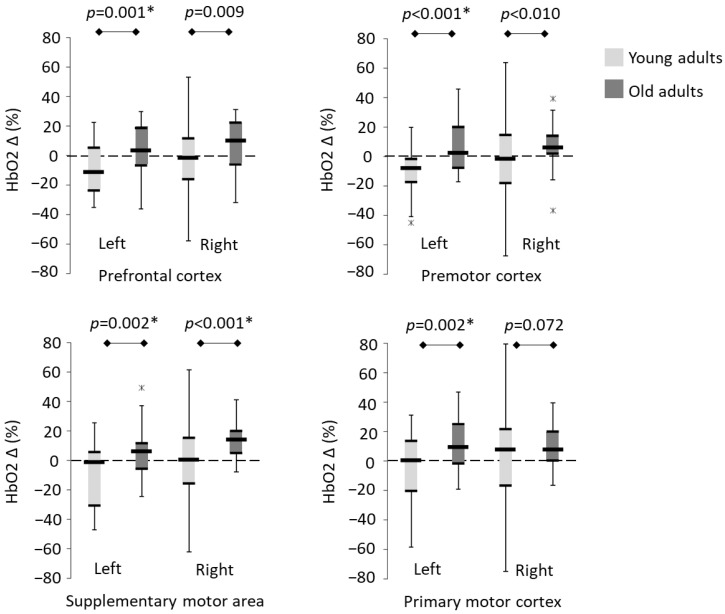
Boxplots depicting regional HbO2 ∆ [fast − preferred] for young and older adults. * denotes significant differences (Bonferroni correction, *p* ≤ 0.006). Individual data values outside 1.5× the interquartile range are highlighted by ‘x’. Error bars reflect minimum and maximum, thin black lines reflect 25th and 75th percentiles and the thick black line reflects the group median.

**Figure 3 sensors-23-06921-f003:**
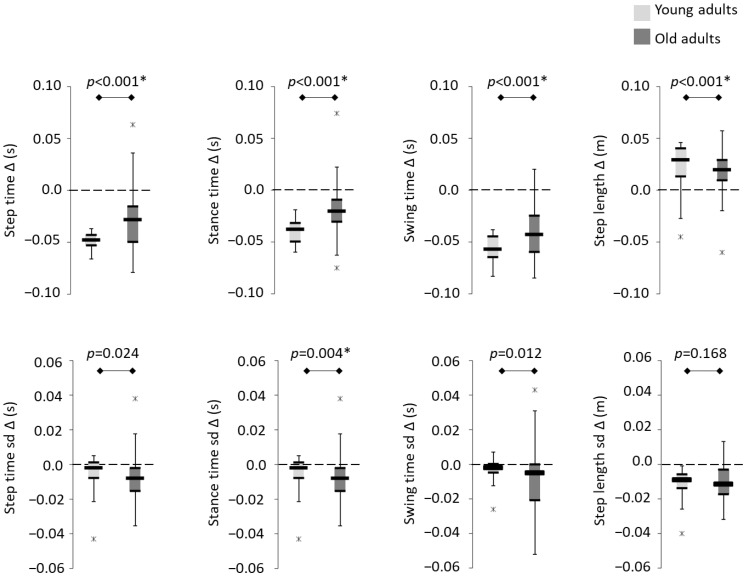
Boxplots depicting temporal-spatial gait outcomes ∆ [fast − preferred] for young and older adults. * denotes significant differences (Bonferroni correction, *p* ≤ 0.006); sd denotes gait variability. Individual data values outside 1.5× the interquartile range are highlighted by ‘x’. Error bars reflect minimum and maximum, thin black lines reflect 25th and 75th percentile and the thick black line reflects the group median.

**Figure 4 sensors-23-06921-f004:**
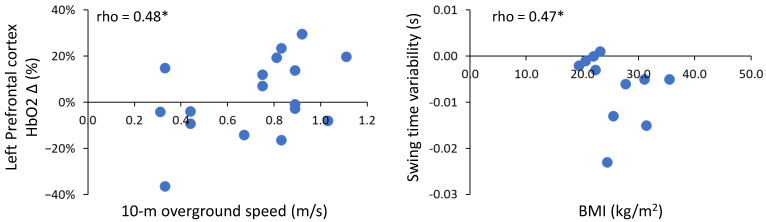
Significant bivariate correlation (rho) between 10 m overground speed and left prefrontal cortex HbO2 in older adults. Blue dots represent individual participants. Δ fast − preferred. Positive values indicate fast > preferred, negative values indicate fast < preferred. * denotes *p* ≤ 0.05.

**Figure 5 sensors-23-06921-f005:**
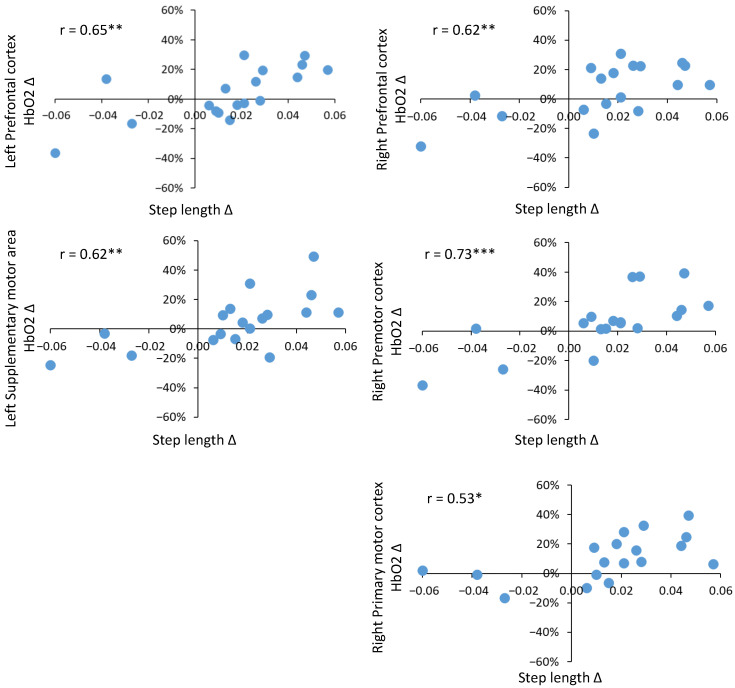
Significant partial correlations between step length (m) and regional HbO2 (%) for older adults. Blue dots represent individual participants. x axis: step length Δ fast − preferred, y axis: regional HbO2 Δ fast − preferred. Positive values indicate fast > preferred, negative values indicate fast < preferred. * denotes *p* ≤ 0.05, ** denotes *p* ≤ 0.01, *** denotes *p* ≤ 0.001.

**Figure 6 sensors-23-06921-f006:**
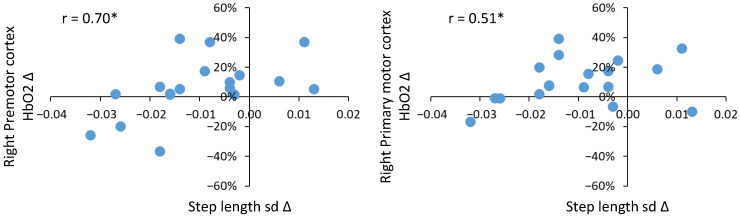
Significant partial correlations between step length variability (m) and regional HbO2 (%) for older adults. Blue dots represent individual participants. x axis: Δ step length fast − preferred, y axis Δ regional HbO2 fast − preferred). Positive values indicate fast > preferred, negative values indicate fast < preferred. * denotes *p* ≤ 0.05.

**Table 1 sensors-23-06921-t001:** Group demographic, anthropometric and clinical descriptors stratified by age.

Descriptors	Young Adults (n = 17)	Old Adults (n = 18)	Sig. (*p*)
Age (y)	20.3 [1.2]	72.6 [8.0]	*p* < 0.001
Sex (f/m)	9 f/ 8 m	9 f/ 9 m	*p* = 0.565 ^a^
Height (m)	1.73 [0.10]	1.69 [0.08]	*p* = 0.274
Mass (kg)	65.5 [13.6]	74.1 [18.6]	*p* = 0.128
BMI (kg/m^2^)	22.0 [20.1, 24.0]	23.5 [21.3, 30.0]	*p* = 0.072 ^b^
Global cognition (MOCA)	28 [1]	28 [1]	*p* = 0.892
Fear of falling (FES-I)	17 [16, 19]	18 [17, 20]	*p* = 0.143 ^b^
10-m overground speed (m/s)	1.08 [0.19]	0.73 [0.25]	*p* < 0.001

Data are reported mean [SD] except for sex (count). BMI and fear of falling are reported median [25th, 75th percentile]. Group differences were evaluated using independent *t*-tests unless stated otherwise; ^a^ Chi squared ^b^ Mann–Whitney U. Sig. denotes significance (*p* ≤ 0.05).

**Table 2 sensors-23-06921-t002:** Paired comparisons evaluating differences in regional HbO2 concentration (%) when walking on a treadmill at a preferred and fast speed.

Cortical Region (HbO2; %)	Young Adults (n = 17)				Old Adults (n = 18)			
	Preferred	Fast	Sig. (*p*)	Effect Size	Preferred	Fast	Sig. (*p*)	Effect Size
Left Prefrontal cortex	−7.3 [32.0]	−11.0 [33.0]	*p* = 0.419	-	−1.8 [31.2]	6.9 [30.8]	*p* = 0.026	-
Left Premotor cortex	−0.6 [41.7]	−5.3 [37.8]	*p* = 0.224	-	−5.1 [31.2]	3.5 [29.9]	*p* = 0.035	-
Left Supplementary motor area	2.2 [27.6]	−2.0 [33.2]	*p* = 0.429	-	−10.2 [35.8]	−2.5 [32.7]	*p* = 0.069	-
Left Primary motor cortex	2.7 [35.0]	3.5 [35.8]	*p* = 0.881	-	−9.6 [32.9]	3.7 [29.6]	*p* = 0.004 *	0.423
Right Prefrontal cortex	−3.3 [32.4]	−3.8 [35.0]	*p* = 0.903	-	−5.4 [30.8]	3.7 [28.0]	*p* = 0.026	-
Right Premotor cortex	3.5 [33.9]	1.5 [35.3]	*p* = 0.659	-	−3.3 [29.8]	5.2 [27.6]	*p* = 0.063	-
Right Supplementary motor area	−3.6 [37.1]	−3.2 [35.0]	*p* = 0.930	-	−5.9 [31.1]	8.7 [29.7]	*p* < 0.001 *	0.480
Right Primary motor cortex	6.2 [32.5]	11.7 [36.9]	*p* = 0.242	-	−10.6 [35.8]	1.5 [32.8]	*p* = 0.002 *	0.352

Pairwise comparisons performed using paired samples *t*-test and regional HbO2; data are presented as mean [SD]; * denotes significant differences (Sig. (*p*)) post-Bonferroni correction (*p* ≤ 0.006); effect sizes are reported for significant comparisons (Hedges g).

**Table 3 sensors-23-06921-t003:** Paired comparisons evaluating differences in temporal-spatial gait outcomes when walking on a treadmill at a preferred and fast speed.

Temporal-Spatial Gait Outcomes	Young Adults (n = 17)				Old Adults (n = 18)			
	Preferred	Fast	Sig. (*p*)	Effect size	Preferred	Fast	Sig. (*p*)	Effect Size
Mean gait outcomes								
Step time (s) ^a^	0.61 [0.05]	0.56 [0.05]	*p* < 0.001 *	0.955	0.61 [0.07]	0.58[0.06]	*p* = 0.005 *	0.416
Stance time (s) ^a^	0.73 [0.04]	0.69 [0.05]	*p* < 0.001 *	0.874	0.74 [0.07]	0.72 [0.06]	*p* = 0.046	
Swing time (s) ^a^	0.49 [0.06]	0.43 [0.06]	*p* < 0.001 *	0.969	0.49 [0.07]	0.45 [0.07]	*p* < 0.001 *	0.580
Step length (m) ^a^	0.52 [0.06]	0.55 [0.06]	*p* = 0.001 *	0.350	0.48 [0.06]	0.49 [0.05]	*p* = 0.053	
Gait variability								
Step time (s) ^b^	0.021 [0.02, 0.03]	0.016 [0.01, 0.02]	*p* = 0.017		0.044 [0.02, 0.15]	0.027 [0.02, 0.08]	*p* = 0.018	
Stance time (s) ^b^	0.026 [0.02, 0.03]	0.022 [0.02, 0.03]	*p* = 0.039		0.052 [0.03, 0.16]	0.033 [0.02, 0.09]	*p* = 0.031	
Swing time (s) ^b^	0.023 [0.02, 0.03]	0.021 [0.02, 0.03]	*p* = 0.039		0.050 [0.02, 0.11]	0.032 [0.02, 0.07]	*p* = 0.078	
Step length (m) ^b^	0.056 [0.04, 0.07]	0.045 [0.03, 0.06]	*p* < 0.001 *	0.612	0.090 [0.05, 0.13]	0.070 [0.04, 0.11]	*p* = 0.006 *	0.467

Pairwise comparisons performed using ^a^ paired samples *t*-test and ^b^ Mann–Whitney U tests. Mean gait outcomes are presented as mean [SD]; gait variability outcomes are presented as median [25th, 75th percentile]. * denotes significant differences (Sig. (*p*)) post-Bonferroni correction (*p* ≤ 0.006); effect sizes are reported for significant comparisons (Hedges g or r as appropriate).

**Table 4 sensors-23-06921-t004:** Relationships between regional HbO2 ∆ [fast − preferred] (%) and temporal-spatial gait ∆ [fast − preferred] for young and older adults.

Young (n = 17)	10 m Walk Speed ^a^	BMI ^a^	LPFC ^b^	LPMC ^b^		LSMA ^b^	LM1 ^b^	RPFC ^b^	RPMC ^b^	RSMA ^b^	RM1 ^b^
10-m gait speed ^a^	-	0.25	−0.08	−0.27		−0.03	0.04	−0.20	−0.03	0.34	0.19
BMI ^a^	0.25	-	0.11	0.01		0.06	0.15	−0.01	0.09	0.09	0.06
Mean											
Step time (s) ^b^	0.20	−0.06	0.04	0.04		−0.11	0.11	0.09	0.07	−0.10	−0.17
Stance time (s) ^b^	0.38	0.33	0.08	−0.13		−0.02	0.18	0.25	0.05	0.06	−0.03
Swing time (s) ^b^	−0.11	−0.27	−0.02	0.18		−0.10	0.00	−0.12	0.07	−0.16	−0.16
Step length (m) ^b^	0.39	0.08	0.28	−0.18		0.06	−0.11	0.42	0.12	−0.06	0.04
Variability											
Step time (s) ^b^	0.14	−0.34	0.17	0.18		0.10	0.03	0.32	0.31	0.19	0.23
Stance time (s) ^b^	0.14	−0.13	0.03	0.19		0.05	0.06	0.17	0.22	0.24	0.17
Swing time (s) ^b^	−0.04	−0.25	0.06	0.10		−0.05	−0.11	0.17	0.07	0.16	0.04
Step length (m) ^b^	0.19	0.31	0.26	−0.03		0.16	−0.07	0.51	0.24	−0.01	0.09
**Old (n = 18)**	**10 m walk speed ^a^**	**BMI ^a^**	**LPFC ^b^**	**LPMC ^b^**		**LSMA ^b^**	**LM1 ^b^**	**RPFC ^b^**	**RPMC ^b^**	**RSMA ^b^**	**RM1 ^b^**
10-m gait speed ^a^	-	0.06	0.48*	0.45		0.36	0.23	0.44	0.33	0.25	0.31
BMI ^a^	0.06	-	0.15	0.26		0.31	0.26	0.06	−0.03	0.12	0.25
Mean											
Step time (s) ^b^	−0.46	0.07	−0.03	−0.09		0.06	0.03	0.19	0.25	0.10	0.03
Stance time (s) ^b^	−0.46	0.04	0.12	−0.05		0.08	0.16	0.31	0.43	0.17	0.18
Swing time (s) ^b^	−0.32	0.12	−0.25	−0.12		−0.01	−0.17	−0.12	−0.01	0.00	−0.24
Step length (m) ^b^	0.36	0.26	0.65 **	0.17		0.62 **	0.46	0.62 **	0.73 ***	0.47	0.53 *
Variability											
Step time (s) ^b^	0.20	−0.20	−0.14	−0.05		−0.24	0.12	0.02	0.11	0.12	0.05
Stance time (s) ^b^	0.38	−0.20	−0.13	0.00		−0.32	0.09	−0.04	0.10	0.13	0.09
Swing time (s) ^b^	0.07	−0.47 *	−0.17	−0.09		−0.33	−0.15	−0.32	−0.01	−0.06	−0.24
Step length (m) ^b^	−0.12	−0.46	0.48	0.16		0.25	0.37	0.45	0.70 *	0.42	0.51 *
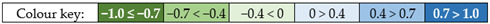

* denotes *p* ≤ 0.05, ** denotes *p* ≤ 0.01, *** denotes *p* ≤ 0.001. ^a^ denotes bivariate correlations (Spearman’s rho) between 10 m overground preferred gait speed and BMI × regional HbO2 and temporal-spatial gait outcomes (Δ; fast − preferred); ^b^ denotes partial correlations regional HbO2 and temporal-spatial gait outcomes (Δ; fast − preferred) controlling for 10 m overground preferred gait speed and BMI. LPFC: left prefrontal cortex, RPFC: right prefrontal cortex, LPMC: left premotor cortex, RPMC: right premotor cortex, LSMA: left supplementary motor area, RSMA: right supplementary motor area, LM1: left primary motor cortex, RM1: right primary motor cortex.

## Data Availability

All data presented within the current study can be obtained from the corresponding author.

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
