# Peer review of "Faster Walking Speeds Require Greater Activity from the Primary Motor Cortex in Older Adults Compared to Younger Adults"

_sensors, 2023, doi:10.3390/s23156921_

Round 1

Reviewer 1 Report

This is a well-conducted study that investigates the relationship between age-related changes in regional cortical activity in the frontal cortex and temporal-spatial gait outcomes during self-selected preferred and fast walking speeds. The research objective is clear and the methodology appears sound. Some of my minor comments are as follows:

-In the methods, including information around the study design will be useful.

-Page 3 Ln107-109: The reason for recruiting more participants beyond the initially estimated sample size of 15 per group is unclear. Additionally, it is not stated why an unequal number of participants were recruited. Were there any dropouts, or was this decision predetermined? Further clarification is needed regarding the rationale behind the recruitment process and any deviations from the anticipated sample size and distribution.

Author Response

sensors-2444393 - Cortical control of walking: influence of age and gait speed.

We thank the reviewers for their careful and thorough review. These valuable insights have considerably improved the manuscript. We have made a number of revisions throughout which we hope addresses the reviewers’ concerns.  Individual responses to each of the reviewers’ comments are provided below. Page (P) and line (L) numbers are included for reference to the location of change within the revised manuscript.

Reviewer 1

This is a well-conducted study that investigates the relationship between age-related changes in regional cortical activity in the frontal cortex and temporal-spatial gait outcomes during self-selected preferred and fast walking speeds. The research objective is clear and the methodology appears sound. Some of my minor comments are as follows:

Thank you for your positive comments.

In the methods, including information around the study design will be useful.

The present study followed the same block design used in our previously published studies (Vitorio et al., 2018 NNR; Stuart et al., 2018 International Journal of Psychophysiology). We referred to this work in our original manuscript (P3, L122).

Following the reviewers’ suggestion, we noted that the full description of the study design and methods including the familiarisation procedures, experimental block design and experimental manipulations were wrongly included in “Section 2.1 Participants”. We have now relocated this information in Section 2.2 which we are updated to: “2.2. Study design, apparatus and data acquisition” (P3, L115-122).

Page 3 Ln107-109: The reason for recruiting more participants beyond the initially estimated sample size of 15 per group is unclear. Additionally, it is not stated why an unequal number of participants were recruited. Were there any dropouts, or was this decision predetermined? Further clarification is needed regarding the rationale behind the recruitment process and any deviations from the anticipated sample size and distribution.

The power calculation established the smallest number of participants necessary using preliminary pilot data collected from healthy younger and older participants. We have further clarified this in the manuscript (P3, L101). We recruited more than 15 participants per group (17 young and 18 older adults) to account for incomplete datasets due to technical difficulties and have clarified this within the manuscript (P3, L104-105).

Reviewer 2 Report

This article is about a NIRS study during two walking tasks (comfortable vs fast) in two groups of participants. Although the topic is of interest, the article requires review in all its parts.

First of all, there are many typos so that some sentences are incomplete. The title perhaps needs to be revised to better reflect the purpose of the article. The work is interesting but in fact it does not present complex measures and conditions from which it is possible to infer about motor control during gait. In the same time it's too long and some sections, like results and discussions, are too verbose. What the salient results are and why, for example, this study is innovative compared to others already present in the literature on the same topic is elusive.

Here are some suggestions for each individual section:

The introduction is well written and articulated. However, it could be integrated with some scientific literature references on locomotor cognitive synchronization tasks.

Some references are suggested:

PMID: 20023000

PMID: 24886198

PMID: 24847239

PMID: 34324958

In the methods, specify that the calculation of the sample size is done taking into account the study cited in reference 50.

For paragraph 2.3.1 Gait , consider inserting the custom algorithm script with which the 8 temporal gait outcomes were extracted. Specify which are the 8 indices used.

From Statistical Analysis onwards the paragraphs and subparagraphs have incorrect numbering.

Please, in subsection 2.3.1 of Statistics, write the formulas in a more readable format and not directly in the text.

Specify what SQRT(N) is. Specify (x-M)/sd) and transcribe the formula in a conventional way, not inserting it in the text.

Specify what LMEM is and all other abbreviations not specified in the text.

In the results section 3.1.3. Gait in young adults, specify which 5 outcomes out of 8 are different. Specify the direction of the difference.

Fig 1 was printed badly, it should be done again in high resolution. Figure legend looks like a paragraph, remove paragraphs and typos (see line 316).

Although one can understand the importance of reporting the results in subparagraphs, the paragraph itself is very dispersive. Authors should try to write the results emphasizing the main effects found. The rest could be included as supplementary material.

In the figures there are many acronyms that are not easy to remember. It would be advisable to always report what the acronyms are. Even the results of the correlations are many and difficult to read. It would be better to move the complete table as supplemetary material and leave only the most relevant results and figures in the text.

Discussion needs to be shortened and should clearly state what are the main findings of this study with respect to the relevant literature.

Author Response

sensors-2444393 - Cortical control of walking: influence of age and gait speed.

We thank the reviewers for their careful and thorough review. These valuable insights have considerably improved the manuscript. We have made a number of revisions throughout which we hope addresses the reviewers’ concerns.  Individual responses to each of the reviewers’ comments are provided below. Page (P) and line (L) numbers are included for reference to the location of change within the revised manuscript.

Reviewer 2

This article is about a NIRS study during two walking tasks (comfortable vs fast) in two groups of participants. Although the topic is of interest, the article requires review in all its parts.

Thank you for your positive comments.

First of all, there are many typos so that some sentences are incomplete. The title perhaps needs to be revised to better reflect the purpose of the article. The work is interesting but in fact it does not present complex measures and conditions from which it is possible to infer about motor control during gait. In the same time it's too long and some sections, like results and discussions, are too verbose. What the salient results are and why, for example, this study is innovative compared to others already present in the literature on the same topic is elusive.

We thank the reviewer for highlighting these errors – we have read through the manuscript and corrected typos throughout.

Thank you for the insightful comments re: the original manuscript title: “Cortical control of walking: influence of age and gait speed”. We have opted to remove the word ‘control’ from the title and amend to: “Faster walking speeds require greater activity of the primary motor cortex in older adults compared to younger adults”

We agree with the reviewer that some sections were overly long. We have streamlined the content and reduced the word count overall.

This study adds new knowledge complementary to existing findings (Suzuki et al., 2004 Neuroimage; Harada et al., 2009 Experimental Brain Research; Lin et al., 2022 Human Movement Science) as it is the first study to evaluate the role of the primary motor cortex in combination with other cortical regions when walking faster, in addition to understanding how this is altered in older age (P2, L86-87). The revised study title now reflects the study novelty.

The introduction is well written and articulated. However, it could be integrated with some scientific literature references on locomotor cognitive synchronization tasks.

Some references are suggested:

PMID: 20023000; PMID: 24886198; PMID: 24847239; PMID: 34324958

Thank you for highlighting these important papers. We have integrated these into the Introduction:

Yogev-Seligmann 2010 P2, L81-82 [REF #49]

Mirelman 2014 P2, L52 [REF #14]

and Discussion:

Kline 2014 P12, L397-398 [REF #77]

De Bartolo 2021 P15, L546-548 [REF #101]

In the methods, specify that the calculation of the sample size is done taking into account the study cited in reference 50.

To ensure that our analyses were sufficiently powered, we computed the power calculation based on using preliminary pilot data collected from healthy younger and older participants. The same power calculated was used in our previously published studies [refs 13 and 50]. This has been clarified within the revised manuscript (P3, L101 and L104).

For paragraph 2.3.1 Gait , consider inserting the custom algorithm script with which the 8 temporal gait outcomes were extracted. Specify which are the 8 indices used.

Thank you for your suggestion. We have summarised the algorithm briefly and provided reference to the original manuscript detailing the algorithm [ref 66 in original manuscript, ref 67 in revised manuscript]. In addition, we have provided a further supporting reference which includes the pseudocode for the gait algorithm used in the present analysis and documented this in the revised manuscript (P4, L170):

Del Din et al., 2016. Instrumented gait assessment with a single wearable: an introductory tutorial. F1000Research. 14;5(2323):2323.

The eight gait outcomes extracted included:

Mean step length (m)

Mean step time (seconds)

Mean stance time (seconds)

Mean swing time (seconds)

Step length variability

Step time variability

Stance time variability

Swing time variability

This information was incorporated within the Methods of our original manuscript (Section 2.3.2. Gait: P4, L171-173).

From Statistical Analysis onwards the paragraphs and subparagraphs have incorrect numbering.

We thank the reviewer for highlighting this error which has now been corrected.

Please, in subsection 2.3.1 of Statistics, write the formulas in a more readable format and not directly in the text.

Specify what SQRT(N) is. Specify (x-M)/sd) and transcribe the formula in a conventional way, not inserting it in the text.

In response to both of the reviewers comments regarding the insertion of equations, we have now distinguished these from the main text and ensured that abbreviations are numbered and cross-referenced within the text (P5, L198-210).

Specify what LMEM is and all other abbreviations not specified in the text.

LMEM denotes linear mixed effects models. We thank the reviewer for highlighting this omission and have amended so that this term is defined upon first use (P5, L212).

In the results section 3.1.3. Gait in young adults, specify which 5 outcomes out of 8 are different. Specify the direction of the difference.

This is included in the manuscript (P5, L260-262).

“When walking fast, the young adults walked with a shorter step time, stance time and swing time (p ≤ 0.001), in-creased step length (p ≤ 0.001) and reduced step length variability, compared to when walking at a preferred speed (Table 3, and Figure 1).”

Fig 1 was printed badly, it should be done again in high resolution. Figure legend looks like a paragraph, remove paragraphs and typos (see line 316).

We apologise that the figure was not of sufficient quality. We have re-exported as a high-quality tiff image and reduced the number of acronyms in the figure for clarity and to reduce the text in the figure footnote.

Although one can understand the importance of reporting the results in subparagraphs, the paragraph itself is very dispersive. Authors should try to write the results emphasizing the main effects found. The rest could be included as supplementary material.

We appreciate the reviewer's comment and have reduced the number of sub-sections and merged shorter, related paragraphs throughout the manuscript – particularly in the Methods and Results sections.

In the figures there are many acronyms that are not easy to remember. It would be advisable to always report what the acronyms are. Even the results of the correlations are many and difficult to read. It would be better to move the complete table as supplemetary material and leave only the most relevant results and figures in the text.

We acknowledge the reviewers concerns and have reduced the number of acronyms where possible (i.e. throughout the main text, Table 2, Figures 1 and 2 and the Supplementary Material). We have opted to retain the abbreviations (for cortical regions only) in the correlation matrix (Table 4) so that the results are legible and may be viewed collectively within a single table. Complete description of these abbreviations accompanies the footnotes beneath the table. We consider the correlation matrix to provide an informative visual representation of the data collectively complementing the results text and figures, such that patterns in the relationships between gait and regional cortical activity are highlighted and as such would like to retain this in the main body of the manuscript.

Discussion needs to be shortened and should clearly state what are the main findings of this study with respect to the relevant literature.

We agree with the reviewer that the Discussion section was long and have refined this section.

Reviewer 3 Report

This paper examines the relationship between cortical activation, as measured with fNIRS, walking speed and age.  The primary goal is to evaluate how walking speed affects cognitive demand in older adults.  It is interesting and well-written. The analysis of the data is extensive but fairly clear in the text.  I congratulate the authors for an excellent paper.  I have a few minor suggestions below for improvement.  

line 77: you mention tDCS both here and in the discussion.  It seems to be unrelated to the current paper.  While I assume that this is something the authors are interested in, I suggest removing this reference in introduction but perhaps mentioning it in a future work section. 

Table 1:  In the significance column there are notations for 'a' and 'b' but I do not see what these labels mean. Please update. 

line 278: Separate heading for 3.1.1. and 3.1.2 is not necessary.  I would combine these.  One sentence for a heading is appropriate. 

Fig. 1: These radar plots are hard to interpret and I am not sure that they are helpful.  Table 2 seems sufficient for presenting the data.  

line 322: I cannot access the supplementary table.  

Table 4: please add more information about the color key in the table notes.  

line 436: Again mention of tDCS should not be here but perhaps in the future studies section.  

line 441: Typo here.  'M1 may serve a as..... " 

Author Response

sensors-2444393 - Cortical control of walking: influence of age and gait speed.

We thank the reviewers for their careful and thorough review. These valuable insights have considerably improved the manuscript. We have made a number of revisions throughout which we hope addresses the reviewers’ concerns.  Individual responses to each of the reviewers’ comments are provided below. Page (P) and line (L) numbers are included for reference to the location of change within the revised manuscript.

Reviewer 3

This paper examines the relationship between cortical activation, as measured with fNIRS, walking speed and age.  The primary goal is to evaluate how walking speed affects cognitive demand in older adults.  It is interesting and well-written. The analysis of the data is extensive but fairly clear in the text.  I congratulate the authors for an excellent paper.  I have a few minor suggestions below for improvement. 

Thank you for your positive comments.

line 77: you mention tDCS both here and in the discussion.  It seems to be unrelated to the current paper.  While I assume that this is something the authors are interested in, I suggest removing this reference in introduction but perhaps mentioning it in a future work section.

We thank the reviewer for raising this – we have extended our conclusions to other rehabilitation techniques beyond tDCS such as peripheral neuromodulation and have relocated this text to the “Study considerations and recommendations for future work” in line with the reviewers suggestion (P19, L556-560).

Table 1:  In the significance column there are notations for 'a' and 'b' but I do not see what these labels mean. Please update.

The annotations are included to denote the statistical test used to identify significant differences and are included in the footnotes accompanying (below) the table (Table 1, P6).

line 278: Separate heading for 3.1.1. and 3.1.2 is not necessary.  I would combine these.  One sentence for a heading is appropriate.

We appreciate the reviewer's comment and have reduced the number of sub-sections and merged shorter, related paragraphs throughout the manuscript – particularly in the Methods and Results sections.

Fig. 1: These radar plots are hard to interpret and I am not sure that they are helpful.  Table 2 seems sufficient for presenting the data. 

The radar plots visually depict the data provided in Table 2. Transforming outcomes with different units of measurement into z scores enables changes across different outcomes to be plotted on the same graph and scale. This enables changes across a number of outcomes to be reviewed simultaneously.

We have added further detail to our description in the Method text to aid interpretation (P5, L207-208).

line 322: I cannot access the supplementary table. 

Sincere apologies for this omission, the Supplementary Material are included in our resubmission.

Table 4: please add more information about the color key in the table notes. 

We provided a colour key below Table 4 in our original manuscript (P11, L361) and have included further detail re: the colour annotations in the results text (P5, L232-234).

line 436: Again mention of tDCS should not be here but perhaps in the future studies section. 

We thank the reviewer for raising this – we have extended our conclusions to other rehabilitation techniques beyond tDCS such as peripheral neuromodulation and have relocated this text to the “Study considerations and recommendations for future work” in line with the reviewers suggestion (P16, L557-561).

line 441: Typo here.  'M1 may serve a as..... "

Thank you for noting this error, we have amended the text.

Round 2

Reviewer 2 Report

I thank the Authors for the time and effort spent in addressing my comments. 

The manuscript was significantly improved from its original submission, therefore I can suggest the Editor to accept it in its present form.